# Black Soybean Improves Vascular Function and Blood Pressure: A Randomized, Placebo Controlled, Crossover Trial in Humans

**DOI:** 10.3390/nu12092755

**Published:** 2020-09-10

**Authors:** Yoko Yamashita, Asuka Nakamura, Fumio Nanba, Shizuka Saito, Toshiya Toda, Junichi Nakagawa, Hitoshi Ashida

**Affiliations:** 1Department of Agrobioscience, Graduate School of Agricultural Science, Kobe University, 1-1, Rokkodai-cho, Nada-ku, Kobe 657-8501, Japan; asukarirarira@gmail.com (A.N.); ashida@kobe-u.ac.jp (H.A.); 2Fujicco Co. Ltd., Research Development, 6-13-4, Minatojima-Nakamachi, Chuo-Ku, Kobe 650–8558, Japan; nanbafumio@gmail.com (F.N.); s-saito@fujicco.co.jp (S.S.); ttoda@mukogawa-u.ac.jp (T.T.); 3Nakagawa Clinic, 3-15-4, Higashisonoda-cho, Amagasaki 661-0953, Japan; junherz@gmail.com

**Keywords:** vascular function, oxidative stress, polyphenol, black soybean, blood pressure

## Abstract

Vascular dysfunction and injurious stimuli such as oxidative stress are closely related to the risk of cardiovascular diseases (CVD). Dietary polyphenols are reported to exert beneficial effects in reducing the risk of CVD. Black soybean has been used as a nutritionally rich food and contains abundant polyphenols in its seed coat and grain. Black soybean has many beneficial physiological activities, and its prevention effects on CVD risk were reported mainly in animal experiments. In this study, we performed a randomized, single blind, placebo controlled, crossover trial to investigate the effect of black soybean consumption on the vascular function in healthy humans. Twenty-two healthy adults aged from 30 to 60 completed the four week trial with daily consumption of about a 40 g test material cookie containing 20 g roasted black soybean powder. Body composition, vascular function, biomarkers for oxidative stress, and polyphenol contents in the urine and the plasma were measured. After ingestion of the black soybean cookie, vascular function, which was evaluated by plethysmogram using a Pulse Analyzer^®^, was improved and systolic blood pressure was decreased. Moreover, nitric oxide levels in plasma and urine were increased, while an oxidative stress biomarker, 8-hydroxy-2′-deoxyguanosine level, in the plasma was decreased accompanied by an increase in the concentration of polyphenols derived from black soybean in plasma and urine. These results suggest that the antioxidant activity of black soybean polyphenols and an increase in the nitric oxide level may contribute to the improvement of vascular function. Thus, black soybean is an attractive food material for improvement of vascular function through decreasing oxidative stress by its potent antioxidant activity and increasing the nitric oxide level in healthy humans.

## 1. Introduction

Maintaining vascular function is important to human health, because vascular dysfunction could be a risk factor of cardiovascular diseases (CVD) [1]. Oxidative stress is a major trigger of vascular dysfunction. It is widely accepted that oxidative stress is closely related to the aging process [2], and decline in vascular function is an inevitable result of aging. Recently, many researchers are focusing on the effect of dietary intake of polyphenols on the prevention of CVD in vivo and in vitro [3,4]. Nitric oxide (NO) is another indicator for regulation of vascular function. It contributes to the relaxation of blood vessels and eventually leads to improvement in vascular stiffness [5]. Oxidative stress and aging quenches the production of NO and hampers the NO-mediated responses that eventually lead to vascular dysfunction [6]. Black soybean is a nutrient-rich food that contains abundant polyphenols in its seed coat and grain and is widely eaten in Eastern Asian countries. It contains anthocyanins and flavan-3-ols including epicatechin, procyanidin B2, procyanidin C1, and cinnamtannin A2 in its seed coat, in contrast to yellow soybean [7,8]. In addition, isoflavones are major polyphenols in the grain of black soybean, as they are in the yellow one. Previous studies reported that polyphenols in the black soybean had various beneficial physiological functions, such as antioxidant in HepG2 cells [9], anti-obesity, and anti-diabetic activities in C57BL/6 mice [10].

Since antioxidant activity of black soybean polyphenols is expected to exert prevention of CVD, many researchers addressed this issue. However, amelioration and prevention effects of black soybean on CVD were mainly reported in animal experiments [11,12,13,14]. Recently, we reported that an intake of roasted black soybean at 30 g/day for 4 and 8 weeks improves vascular function and vascular age through increasing NO production and reducing oxidative stress in healthy women by an open-label trial [15]. Moreover, we found that black soybean seed coat extract increased the NO production accompanied by Glucagon-like peputide-1 (GLP-1) secretion from the intestine of rats [16]. Another research group demonstrated that consumption of black soybean (35% of experiment diet) for 10 weeks inhibited oxidative stress by increasing antioxidant activity and improving lipid profiles in ovariectomized rats, resulting in the improvement of risk factors associated with CVD [11]. It was also demonstrated that an oral administration of black soybean extract at 50 and 100 mg/kg body weight for 14 days reduced the risk of CVD by improving blood circulation through inhibiting platelet aggregation and thrombus formation in rats [12]. From these reports, it can be expected that black soybean improves vascular function and prevents CVD.

However, there is not enough evidence for the improvement of vascular function in humans after consumption of black soybean. As mentioned above, we reported that consumption of roasted black soybean improves vascular function in healthy women, but there are two remaining issues. One is that previous participants were only woman, and another is the design of study. In the previous study, all participants were aware of the treatment because of the open-label trial without a placebo meal. To elucidate the exact effects of black soybean on the improvement of vascular function, in this study, we performed a randomized, placebo-controlled, crossover trial in both healthy men and women using a test cookie containing black soybean powder and a placebo cookie. In this trial, the ingestion amounts of black soybean were reduced from 30 g to 20 g, and the period was shortened from 8 weeks to 4 weeks; we also investigated vascular function, production of nitric oxide, oxidative stress, and polyphenol concentrations in the plasma and the urine.

## 2. Materials and Methods

### 2.1. Participants Eligibility

This study was approved and conducted by the Institutional Review Board of Fujicco Co., Ltd. (randomized controlled trials registration number: #5801) in accordance with the Declaration of Helsinki. To perform the study, informed consent was obtained from all participants. Participants were excluded if they met any of the following exclusion criteria: (1) having a clinical history of severe gastrointestinal disease, liver disease, kidney disease, or heart disease; (2) currently undergoing treatment for metabolic syndrome or its associated diseases; (3) using medication for blood flow and/or pressure, such as Warfarin and Captopril; and (4) having a physical condition considered inappropriate for this study by a physician. The main inclusion criterion was that the vascular age of the participant was higher than their chronological age.

### 2.2. Design of Human Study

This study was a randomized, placebo-controlled, crossover trial. The required sample size assumed 34 participants (17 per group) with a power of 80%, a two-tailed α level of 0.05, and a 50% treatment effect by G*power free software [17]. From this assumption, 23 participants (12 males and 11 females) aged from 30 to 60 years were enrolled for this study. Participants were randomly allocated to receive daily about 40 g/day of cookie containing either 20 g of roasted black soybeans powder or 20 g of flour as a placebo for 4 weeks, with a washout period of 4 weeks between the treatments. In the previous study [15], participants consumed 30 g of roasted black soybean (whole grain) as the test meal for 4 and 8 weeks, and consumption of black soybean markedly improved the vascular function after 4 weeks. Therefore, in the present study, we reduced the intake amount of black soybean to 20 g and decided on a test period of 4 weeks. Since polyphenols are recognized as xenobiotics, they are easily excreted into urine and feces by 24 h [18]. In addition, many crossover trials employ a 4 weeks washout period [19]. Thus, we also decided on the 4 weeks washout period in the present study. At the beginning and the end of each treatment, anthropometrics, accelerated plethysmogram (APG), and blood pressure and assessments of health-related quality of life (HRQOL) were measured; blood and urine samples were also collected from participants under the fasting condition before breakfast.

### 2.3. Study Diets

Test cookies were produced by Fujicco Co., Ltd. (Kobe, Japan) and Bakery Miki. The composition of the cookies is shown in Table 1A, and their nutrients composition was calculated and is shown in Table 1B. The used black soybean cultivar was “Hikariguro”. Indigestible dextrin was used to form the cookie (Pine Fiber W, Matsutani Chemical Industry Co., Ltd., Itami, Japan). Nutritional composition of test cookies was calculated using the Standard Tables of Food Composition in JAPAN, 2015. Polyphenol contents of each test cookie were also measured using the same methods as described in the previous report [15]. In brief, a cookie was pulverized in a mortar, and the powdered cookie was washed by hexane to remove fat. After removing the hexane layer, the residue was extracted with 70% acetone containing 0.5% acetic acid. The acetone extract was provided for the measurement of polyphenol contents by a high-performance liquid chromatography (HPLC). The detailed method is written in the later section (Section 2.8, Extraction and Quantification of Polyphenols.) For the antioxidant activity, the 2, 2′-azobis (2-amidinopropane) dihydrochloride (AAPH) radical absorbance capacity of black soybean and placebo cookies was measured [9].

### 2.4. Measurements of Body Composition

Body composition, body weight, body mass index (BMI), body fat percentage, visceral fat percentage, biological age, basal metabolic rate, estimated bone mass, and muscle mass were measured by a Bioelectrical Impedance Analysis using a body composition meter (BC-610-PB, TANITA. Co., Ltd. Tokyo, Japan).

### 2.5. Measurements of Vascular Function

Participants took a rest of about 10 min, and then vascular function and blood pressure were measured. Vascular function was evaluated by acceleration plethysmogram (APG) using a Pulse Analyzer^®^ device (Pulse Analyzer Plus View^®^, YKC Corporation, Tokyo, Japan) as described in the previous report [15]. Briefly, APG was measured with the device attached to the left middle finger of each participant. APG is the second derivative wave of the photoplethysmogram, which consisted of *a*, *b*, *c,* and *d* waves, namely, early systolic positive wave, early systolic negative wave, late systolic re-increasing wave, and late systolic re-decreasing wave, respectively. Their magnitudes and the height ratios of b/a, c/a, and d/a were measured. Vascular age was calculated from the second derivative of the photoplethysmogram aging index, which is [(b-c-d-e)/a] wave ratio [20,21,22]. Vascular waveform, waveform score, and peripheral vascular health were calculated from a wave pattern, and the ratios of these 4 waves were calculated using software for the Pulse Analyzer^®^. Systolic, diastolic, and central blood pressure were measured in the right upper arm using an automated sphygmomanometer (HEM-9000AI, OMRON Corporation, Kyoto, Japan).

### 2.6. Measurements of Biomarkers in Blood and Urine

Plasma and urine were used for measurement of NO_2_/NO_3_, 8-hydroxy-2′-deoxyguanosine (8-OHdG), hexanoyl-lysine (HEL), and myeloperoxidase (MPO) by corresponding commercial kit [NO: NO_2_/NO_3_ Assay Kit-C II (DOJINDO LABORATORIES, Kumamoto, Japan); 8-OHdG: New 8-OHdG Check ELISA (Japan Institute for the Control of Aging, NIKKEN SEIL Co., Ltd. (JaICA) Shizuoka, Japan); HEL: HEL ELISA kits (JaICA); and MPO: Human serum MPO and urine MPO ELISA kits (JaICA)]. Urinary NO_2_/NO_3_ level was corrected by creatinine equivalent, which was measured by using Creatinine (urinary) Colorimetric Assay Kit (Cayman CHEMICAL, Ann Arbor, MI, USA).

Analyses of hematologic biochemical markers including creatinine, total protein, blood urea nitrogen, glucose, lactate dehydrogenase, alkaline phosphatase, γ- glutamyltranspeptidase, aspartate aminotransferase, alanine aminotransferase, triglycerides, high-density lipoprotein cholesterol, low-density lipoprotein cholesterol, Na^+^, Cl^−^, K^+^, white blood cells, red blood cells, hemoglobin, hematocrit, mean corpuscular volume, mean corpuscular hemoglobin, mean hemoglobin concentration, and platelet were measured by LSI Medience Co. (Tokyo, Japan).

### 2.7. Extraction and Quantification of Polyphenols

Extraction and quantification of polyphenols were performed by the same method as described in our previous report [15,23]. An aliquot of 10 mL urine was concentrated to 2 mL using a centrifugal separator in vacuo. Concentrated urine or 500 μL of plasma were mixed with 2% (w/v) ascorbic acid (200 μL for urine and 50 μL for plasma) to prevent oxidation during extraction and were transferred to polypropylene centrifuge tubes (15 mL, BD Biosciences, San Jose, CA, USA), which were siliconized by Sigmacote^®^ (Sigma-Aldrich, St. Louis, MO, USA) before use. These plasma and urine samples were hydrolyzed with 500 U of β-glucuronidase from *Escherichia coli* (type IX-A, Sigma-Aldrich) and 10 U of sulfatase from *Abalone entrails* (type VIII, Sigma-Aldrich) for deconjugation according to our previous reports [15]. A solid phase extraction method was used to extract polyphenols from the mixture: C18 Sep-Pak cartridge (50 mg resin, Waters Co., Milford, MA, USA) was conditioned with 5 mL of methanol and 5 mL of ultrapure water. Plasma and urine samples were centrifuged at 3000× *g* for 15 min to remove precipitated protein and applied to the cartridge. After the cartridge was washed with 5 mL of 10% methanol, polyphenols were eluted with 2 mL of 95% (v/v) methanol and evaporated to dryness using the centrifugal separator. Obtained extracts were used for analysis of polyphenols by HPLC.

HPLC was performed using a system equipped with a DGU-20A 3R degas unit, LC-20AD XR binary pump, SIL-20AC XR auto sampler, RF-20A XS fluorescence detector, SPD-M20A diode array detector, CTO-20AC column oven, and CBM-20A communications bus module connected to an LC work station (Shimadzu Corporation, Kyoto, Japan). The analytical column was a Cadenza CL-C18 column (φ 250 mm × 4.6 mm, 3 μm, Imtakt, Kyoto, Japan) protected by a guard column (Cadenza CL-C18, φ 5 mm × 2 mm, 3 μm, Imtak). Quantification of cyanidin-3-*O*-glucoside (C3G), flavan-3-ols, and isoflavones was separately carried out under the same analytical conditions with corresponding authentic compound as described in our previous report [15].

### 2.8. Assessments of HRQOL

HRQOL was assessed using the Japanese version of the 36-item Short-Form Health Survey (SF-36) questionnaire developed by iHope International Co., Ltd. (Kyoto, Japan) [24]. SF-36 was used as a profile-type evaluation of HRQOL. In this study, the Japanese version 1.20 was used; the reliability and the validity of the Japanese version of SF-36 have been confirmed in normal subjects [25]. SF-36 can comprehensively evaluate physical and mental functions and can also assess social activities. In this study, eight subscales of SF-36 (physical functioning, role physical, bodily pain, general health, vitality, social functioning, role emotional, and mental health) were used to investigate health-related quality of life.

### 2.9. Statistical Analysis

Data are expressed as the means ± standard deviation. Statistical analysis was performed with a Welch’s *t* test using JMP statistical software version 11.2.0 (SAS Institute, Cary, NC, USA). The level of significance was set as *p* < 0.05.

## 3. Results

### 3.1. Nutritional Composition, Polyphenol Content, and Antioxidant Capacity of Test Cookies

The detailed nutritional composition and the polyphenol contents of each test cookie are shown in Table 1B and Table 2, respectively. Each test cookie had almost the same energy amount of about 200 kcal (Table 1B). The black soybean cookie contained flavan-3-ols, cyanidin-3-*O*-glucoside, and isoflavones, whereas the placebo cookie did not contain these polyphenols (Table 2). The antioxidant activity of each cookie was measured by AAPH radical absorbance capacity method [9]. The antioxidant activity of the black soybean cookie was much higher than that of the placebo cookie. The AAPH radical absorbance capacities of these cookies were 24.6 ± 0.23 and 0.98 ± 0.03 mg Trolox eq/portion, respectively.

### 3.2. General Characteristics of Participants

Before the trial, an average vascular age of all participants was 52.4 ± 1.6 years old with the average BMI of 23.1 ± 0.8 kg/m^2^. During the trial, one participant dropped out for non-health related reasons, and 22 participants completed the 4 week trial with no report on health problems or any abnormality. At the end of the trial, body weight, BMI, and body fat percent did not alter in the black soybean group, but these anthropometric parameters significantly increased in the placebo group (Table 3 and Appendix A). No significant change in hematologic parameters was observed throughout the trial (data not shown). HRQOL was assessed by SF-36, but there was no significant change (data not shown).

### 3.3. Vascular Function

After 4 weeks intake of the black soybean cookie, vascular function was significantly improved (Table 4 and Appendix A). In the black soybean group, vascular age became about 3 years younger at the end of the trial (49.3 ± 9.3 years old) compared to that at the beginning of the trial (52.3 ± 10.2 years old). In addition, the vascular age of the black soybean group was significantly younger than that of the placebo group at the end of the trial (49.3 ± 9.3 years old vs. 52.2 ± 11.2 years old). After the intake of the black soybean cookie, lowered vascular age was observed in 14 of 22 participants whose vascular ages were higher than their chronological age before the trial. On the other hand, the vascular age in the placebo group did not change during the trial. The black soybean group significantly improved vascular waveform, waveform score, and peripheral vascular health compared to those of the placebo group at the end of the trial. The results of blood pressure are also shown in Table 4 and Appendix A. In the black soybean group, the systolic blood pressure significantly decreased at the end of the trial compared to that at the start of the trial (from 129.4 ± 3.9 mmHg to 121.9 ± 3.1 mmHg), and central blood pressure tended to decrease at the end of the trial without statistical significance [from 134.2 ± 4.3 mmHg to 127.6 ± 3.2 mmHg (*p* = 0.065), respectively]. The systolic blood pressure in the black soybean group also significantly decreased compared to that of the placebo group at the end of the trial. On the other hand, blood pressure in the placebo group tended to increase at the end of the trial.

### 3.4. Oxidative Stress Markers in Plasma and Urine

In this study, three oxidative stress markers, namely, 8-OHdG, HEL, and MPO, were selected and measured (Table 5 and Appendix A). Among them, the intake of black soybean cookie significantly decreased the 8-OHdG level in the plasma and tended to decrease it in the urine (*p* = 0.098) at the end of the trial compared to that at the start of the trial. At the end of the trial, the 8-OHdG level in the black soybean group also significantly decreased in plasma and urine compared to that in the placebo group. In the placebo group, these oxidative stress markers did not change during the trial.

### 3.5. NO Concentration in the Plasma and Urine

Since NO is involved in the regulation of vascular function, including blood pressure and blood flow [26], we measured the NO_2_/NO_3_ concentration in the plasma and the urine to estimate the NO production (Table 5 and Appendix A). At the end of the trial, NO_2_/NO_3_ concentration in the black soybean group significantly increased in plasma and urine compared to that in the placebo group. In addition, NO_2_/NO_3_ concentration in the black soybean group tended to increase at the end of the trial compared to that at the start of the trial (*p* = 0.076).

### 3.6. Polyphenol Concentrations in Plasma and Urine

Concentration of polyphenols in plasma and urine was measured by HPLC with and without enzymatic hydrolysis after consumption of test cookies. Results of the polyphenol concentration in plasma and urine are shown as an aglycone and the conjugated forms in Figure 1 and Figure 2, respectively. Supporting data for these figures are shown in Appendix A. Cyanidin-3-*O*-glucoside was detected in the urine but not in the plasma. The aglycone form of cyanidin-3-*O*-glucoside in the urine significantly increased in the black soybean group at the end of the trial compared to that at the start of the trial (Figure 2).

Among flavan-3-ols, only (−)-epicatechin was detected in the plasma. At the end of the trial, both (−)-epicatechin aglycone and its conjugate forms in the black soybean group significantly increased to 1.6- and 3.0- fold, respectively, while those in the placebo group did not change (Figure 1 and Appendix A). On the other hand, four flavan-3-ols, namely, (−)-epicatechin, procyanidin B2, procyanidin C1, and cinnamtannin A2, were detected in the urine (Figure 2 and Appendix A). In particular, procyanidin B2 aglycon and its conjugate forms in the black soybean group significantly increased at the end of the trial compared to those at the start of the trial. At the end of the trial, conjugate forms of procyanidin B2 significantly increased in the black soybean group compared to that in the placebo group. (−)-Epicatechin and procyanidin C1 also showed the same trend without statistical significance. From these results, major polyphenols in the black soybean seed coat were absorbed into the body and then excreted and accumulated in the urine.

Isoflavones are major polyphenols in the soybean grain, and black soybean also contains isoflavones abundantly. Three types of isoflavones (aglycone and corresponding glycoside form) and (*S*)-equol, a metabolite of daidzein, were measured. Daidzein, genistein, glycitein, and (*S*)-equol were detected in the plasma, but isoflavone glycosides were not detected throughout this trial (Figure 1 and Appendix A). Aglycon form of daidzein and (*S*)-equol and conjugate forms of daidzein and genistein in the black soybean group significantly increased at the end of the trial compared to those at the start of the trial. At the end of the trial, they also significantly increased against the placebo group. As shown in Figure 2 and Appendix A, all isoflavones were detected in the urine. Surprisingly, isoflavone glycosides were also detected in addition to their aglycones. Both aglycone and conjugate forms of daidzin, daidzein, genistein, and glycitein in the black soybean group significantly increased at the end of the trial compared to those at the start of the trial. In addition, aglycon form of genistin and glycitin also significantly increased. In the case of (*S*)-equol, its conjugate form tended to increase without significant difference due to large deviation. At the end of the trial, it was noteworthy that all aglycones and conjugate forms of isoflavone, except genistin conjugates, significantly increased in the urine of the black soybean group compared to those in the placebo group.

(*S*)-Equol is a major metabolite of the isoflavone daidzein and has strong biological activity [27]. It is known that there are equol producers and non-producers. In this study, plasma and urine were collected four times from each participant during the trial. Average concentration of (S)-equol was calculated: plasma and urine concentrations of (S)-equol ranged from 3.3–149.2 pM and 0.18–47.3 nM, respectively, with a wide variation. After the consumption of black soybean cookie, plasma and urine concentrations of (S)-equol increased and ranged from 13.0–348.2 pM and 0.22–70.8 nM, respectively.

## 4. Discussion

Recently, we reported that intake of black soybean improved vascular function in healthy women by an open-labeled trial. This study expanded the human trial from the previous study to confirm the intake of black soybean improved vascular function. In the present study, we conducted and performed the randomized, single blind, placebo controlled, crossover trial for 4 weeks by reducing the intake amount of black soybean from 30 g to 20 g. We confirmed that the improvement of vascular function, including lowered vascular age and decreased blood pressure (Table 4 and Appendix A), was accompanied by increased concentration of polyphenols derived from black soybean (Figure 1 and Figure 2 and Appendix A). Obtained results in the current study support our previous one. Moreover, smaller intake amount (20 g of black soybean/day) is enough to improve the vascular function regardless of gender. Therefore, daily intake of black soybean is possible to improve vascular stiffness and to reduce the risk of CVD.

Oxidative stress is closely related to the underlying mechanism of vascular dysfunction. In this study, we used three biomarkers, 8-OHdG, HEL, and MPO. These biomarkers reflect the different aspects of the oxidative stress: 8-OHdG is a marker for oxidative DNA damage [28]; HEL is one of the lipid peroxidation markers [29]; and MPO is another biomarker for lipid peroxidation and inflammation [30]. Of these, 8-OHdG level specifically decreased in plasma and urine (Table 5 and Appendix A), suggesting that the amelioration of oxidative DNA damage would contribute to the improvement of vascular function, because oxidative DNA damage is a trigger of the development of CVD [31]. Our previous report demonstrated that black soybean seed coat polyphenols prevented 2,2′-azobis(2-methylpropionamide) dihydrochloride (AAPH)-induced formation of 8-OHdG in HepG2 cells [9]. It was also reported that drinking red wine suppressed 8-OHdG formation in humans with a Mediterranean diet and an Occidental diet for 3 months [32]. These results indicate that polyphenols would contribute to the improvement of vascular function. In our previous human trial, we found 8-OHdG and HEL levels significantly decreased in the plasma [15]. In the current study, we did not observe the decreasing HEL level. This discrepancy may be due to the difference of the intake amount of black soybean. Taken together, these results show black soybean polyphenols have a potential to decrease oxidative stress markers, and 8-OHdG is the most sensitive biomarker for improvement of vascular function through suppressing oxidative stress.

In this study, we measured 11 black soybean polyphenols and one intestinal bacterial metabolite of daidzein, namely, (*S*)-equol, in the plasma and the urine by HPLC and found epicatechin, daidzein, and genistein in the plasma (Figure 1 and Appendix A) and C3G, procyaniding B2, daidzin, genistin, glycitin, daidzein, genistein, and glycitein in the urine (Figure 2 and Appendix A). These results are reasonable, because it is known that bioavailability of epicatechin and isoflavones is relatively higher than that of C3G and procyanidins [33,34], indicating that epicatechin and isoflavones in black soybean are considerable active compounds for the improvement of vascular function though possessing their antioxidant activity. It is known that soy isoflavones improves vascular functions [35,36]. Interestingly, certain amounts of isoflavones were detected as their aglycone forms in plasma and urine in this study. Usually, isoflavones are contained in soybean as the glycoside forms. This result coincides with those in our previous human trial [15]. Another study also demonstrated the same metabolic conversion after the intake of soybean [37]. Thus, isoflavone glycosides underwent hydrolysis in the intestines and formed aglycones and contributed to the improvement of vascular function. It is reported that (*S*)-equol affects vascular function [38]. In this study, only aglycine form of (*S*)-equol slightly increased with statistical significance in the plasma after consumption of black soybean cookie (Figure 1 and Appendix A). It is therefore suggested that contribution of (*S*)-equol to the improvement of vascular function might be limited. Although active compound was unclear in this study, the results from our previous study demonstrated that monomer to tetramer procyanidins from black soybean reduced 8-OHdG and oxidative stress levels to the same degree [9]. It was therefore suggested that these polyphenols coordinately acted for the improvement of vascular function. Further study is needed to clarify the active compound in the future.

NO is known to regulate vascular function through the relaxation of blood vessels and eventually leads to improvement in vascular stiffness [5]. Oxidative stress decreases the production of NO and its functions, resulting in induction of the vascular dysfunction [6]. In this study, NO_2_/NO_3_ concentration in plasma and urine significantly increased in the black soybean group compared to that in the placebo group. In our previous open-label trial, consumption of black soybean significantly increased the urinary NO_2_/NO_3_ level after 4 and 8 weeks. Recently, we found that flavan-3ols in black soybean promoted NO production through activation of endothelial nitric oxide synthase [16]. Isoflavones and equol were also reported to promote NO production [35,39,40]. These results indicated that consumption of black soybean improved vascular function and blood pressure through increasing NO concentration in addition to antioxidant activity of polyphenols in humans.

Components other than polyphenols in the black soybean may also contribute to the observed effects. It was reported that soy protein, accounting for approximately 36% of dry soybeans by weight, was reported to reduce the risk of CVD through regulating the lipid profile, including lowering total cholesterol, low-density lipoprotein, and triglycerides without affecting high-density lipoprotein [41]. Dietary fiber also reportedly caused a reduction in the blood pressure and improved endothelial function [42]. Actually, the nutrient content of soybean cookie was quite different from that of the placebo cookie in the present study. These nutrients and polyphenols might coordinately play a role in the improvement of vascular function. Therefore, black soybean is an attractive functional food for prevention and/or improvement of vascular function. Daily intake of black soybean may contribute to maintaining human health.

## 5. Conclusions

The intake of a test cookie containing 20 g/day of roasted black soybean powder for 4 weeks significantly improved vascular function and reduced oxidative stress. Polyphenols derived from black soybean also increased in plasma and urine, which contributed to the improvement of vascular function. Thus, black soybean is an attractive food for lowering the risk of CVD in humans.

## Figures and Tables

**Figure 1 nutrients-12-02755-f001:**
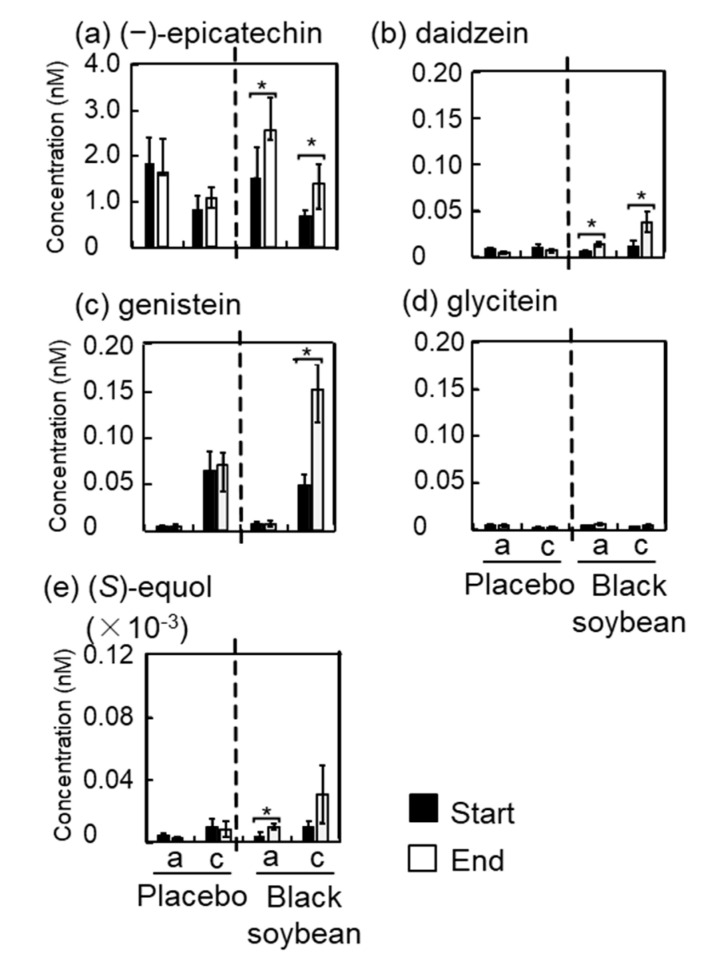
The effect of black soybean consumption on polyphenol concentration in the plasma; a: aglycone form, c: conjugates form. Each panel showed (**a**) (−)-epicatechin, (**b**) daidzein, (**c**) genistein, (**d**) glycitein, and (**e**) (S)-equol, respectively. The results are represented as the means ± standard deviation. * *p* < 0.05 vs. start by Welch’s *t* test.

**Figure 2 nutrients-12-02755-f002:**
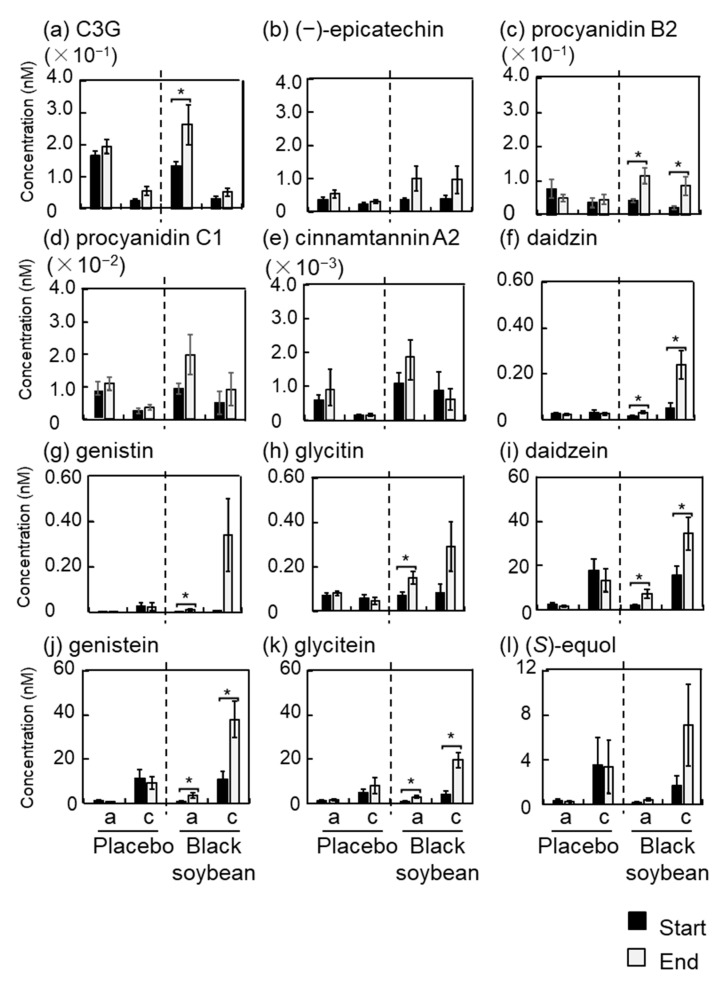
The effect of black soybean consumption on polyphenol concentration in the urine. a; aglycone form, c; conjugates form. Each panel showed (**a**) C3G, (**b**) (−)-epicatechin, (**c**) procyanidin B2, (**d**) procyanidin C1, (**e**) cinnamtanninA2, (**f**) daidzin, (**g**) genistin, (**h**) glycitin, (**i**) daidzein, (**j**) genistein, (**k**) glycitein, and (**l**) (S)-equol, respectively. The results are represented as the means ± standard deviation. * *p* < 0.05 vs. start by Welch’s *t* test.

**Table 1 nutrients-12-02755-t001:** (**A**) Composition of test cookies, (**B**) nutritional composition of test cookies.

(**A**)		
	**Content (g)**	
	**Placebo**	**Black Soybean**
Wheat flour	20	0
Roasted black soybean powder	0	20
Vegetable oil	8.3	7.3
Water	6.1	6.1
Liquid sucrose	5.3	5.3
Sugar	2.3	2.3
Indigestible dextrin	0.1	0.1
Salt	0.1	0.1
Baking powder	4	7.3
Total	46.3	48.5
Weight after baking	39	42
(**B**)		
	**Placebo**	**Black Soybean**
Calories	201.7 kcal	192.5 kcal
Protein	2.0 g	7.5 g
Fat	8.9 g	9.7 g
Carbohydrate	28.4 g	18.8 g
Ash	0.2 g	1.1 g
Sodium	49.6 mg	54.2 mg
Dietary fiber	0.6 g	3.5 g
Salt equivalents	0.1 g	0.1 g

Nutritional composition of test cookies was calculated using the Standard Tables of Food Composition in JAPAN, 2015.

**Table 2 nutrients-12-02755-t002:** Polyphenol contents in test cookies.

	Content (mg)	
	Placebo	Black Soybean
Anthocyanidin		
Cyanidin-3-*O*-glucoside	ND	1.12
Flavan-3-ols		
(−)-Epicatechin	ND	0.038
Procyanidin B2	ND	0.049
Procyanidin C1	ND	0.064
Cinnamtannin A2	ND	0.044
Isoflavones		
Daizein	ND	0.046
Daidzin	ND	5.99
Glycitein	ND	0.04
Glycitin	ND	0.324
Genistein	ND	0.085
Genistin	ND	2.27
Total polyphenol (mg gallic acid equivalent)	3.20 ± 0.02	20.0 ± 3.99
Antioxidant activity (mg Trolox equivalent)	0.98± 0.03	24.6 ± 0.23

Total polyphenol and antioxidant activity were performed in triplicate. Means ± standard deviation are shown. Measurement of each polyphenol was a single analysis. ND: not detected.

**Table 3 nutrients-12-02755-t003:** Anthropometric parameters of participants during the 4 week trial.

	Placebo		Black Soybean	
Anthropometric parameters	Start	End	Start	End
Body weight (kg)	63.8 ± 2.7	64.3 ± 2.7 *	64.0 ± 2.8	63.9 ± 2.7
BMI	23.1 ± 0.8	23.2 ± 0.8 *	23.1 ± 0.8	23.1 ± 0.8
Body fat (%)	23.9 ± 1.7	25.4 ± 1.5 *	24.3 ± 1.7	24.3 ± 1.7
Visceral fat (%)	8.4 ± 1.0	8.4 ± 1.0	8.5 ± 1.0	8.5 ± 1.0
Biological age	36.3 ± 2.8	37.3 ± 2.8	37.1 ± 2.7	37.0 ± 2.8
Basal metabolic rate (kcal/day)	1368 ± 53	1360 ± 52	1366 ± 52	1362 ± 51
Estimated bone mass (kg)	2.6 ± 0.1	2.6 ± 0.1	2.6 ± 0.1	2.6 ± 0.1
Muscle mass (%)	45.7 ± 2.0	45.1 ± 2.0	45.7 ± 2.0	45.5 ± 2.0

Means ± standard deviation are shown. * *p* < 0.05 vs. Start by Welch’s *t* test. BMI: body mass index.

**Table 4 nutrients-12-02755-t004:** The effects of black soybean consumption on the vascular function.

Vascular Function	Placebo		Black Soybean	
	Start	End	Start	End
APG				
Vascular age	52.5 ± 11.0	52.2 ± 11.2	52.2 ± 10.2	49.3 ± 9.3 ^†,^*
*a* wave	109.0 ± 10.9	113.0 ± 7.8	111.1 ± 9.5	109.3 ± 9.1
*b* wave	−57.8 ± 16.2	−59.0 ± 14.7	−58.6 ± 13.1	−62.3 ± 12.9
*c* wave	−32.2 ± 17.9	−36.3 ± 20.7	−35.9 ± 19.3	−27.3 ± 20.4
*d* wave	−50.4 ± 19.3	−55.1 ± 23.3	−56.1 ± 22.9	−44.1 ± 20.5 *
Vascular waveform	3.8 ± 1.4	3.8 ± 1.3	3.8 ± 1.4	3.1 ± 1.3 ^†^
Waveform score	44.5 ± 12.8	43.8 ± 11.6	43.5 ± 12.6	51.0 ± 14.0 ^†,^*
Peripheral vascular health	62.0 ± 12.9	60.6 ± 11.9	62.7 ± 11.4	70.0 ± 15.4 ^†,^*
Blood pressure				
Systolic blood pressure	122.3 ± 3.5	125.7 ± 3.1	129.4 ± 3.9	121.9 ± 3.1 ^†,^*
Diastolic blood pressure	81.0 ± 2.2	82.5 ± 2.3	84.5 ± 2.7	80.9 ± 2.0
Central blood pressure	129.4 ± 3.5	132.6 ± 3.3	134.2 ± 4.3	127.6 ± 3.2

Means ± standard deviation are shown. * *p* < 0.05 vs. start, ^†^
*p < 0.05* vs. end of placebo by Welch’s *t* test. APG: accelerated plethysmogram.

**Table 5 nutrients-12-02755-t005:** The effects of black soybean consumption on the antioxidant activity.

Vascular Function	Placebo		Black Soybean	
	Start	End	Start	End
Plasma				
NO_2_/NO_3_ (μM)	29.1 ± 3.8	9.6 ± 1.4	29.1 ± 2.7	35.4 ± 2.9 ^†^
HEL (nM)	4.3 ± 0.2	4.6 ± 0.2	4.3 ± 0.2	4.5 ± 0.2
MPO (ng/mL)	88.1 ± 3.0	90.6 ± 3.7	86.6 ± 3.0	85.5 ± 2.8
8-OHdG (ng/mL)	1.6 ± 0.1	1.5 ± 0.1	1.5 ± 0.1	1.2 ± 0.1 *^,†^
Urine				
NO_2_/NO_3_ (μM)	20.9 ± 5.3	15.5 ± 3.4	18.8 ± 4.2	31.5 ± 7.2 ^†^
HEL (nM)	110.5 ± 30.0	122.5 ± 41.1	128.5 ± 23.0	114.6 ± 15.6
MPO (ng/mL)	10.5 ± 0.4	14.0 ± 3.9	10.1 ± 0.1	0.1 ± 0.1
8-OHdG (ng/mL)	9.4 ± 1.7	12.5 ± 2.6	8.8 ± 1.5	7.6 ± 0.7 ^†^

Means ± standard deviation are shown. * *p* < 0.05 vs. start, ^†^
*p < 0.05* vs. end of placebo by Welch’s *t* test. NO: nitric oxide; HEL: hexanoyl-lysine; MPO: myeloperoxidase; 8-OHdG: 8-hydroxy-2′-deoxyguanosine.

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
