# Peer review of "Black Soybean Improves Vascular Function and Blood Pressure: A Randomized, Placebo Controlled, Crossover Trial in Humans"

_nutrients, 2020, doi:10.3390/nu12092755_

Round 1
Reviewer 1 Report
Overall comments: The authors conducted a randomized, single-blind, placebo-controlled, crossover trial to test the effects of 20g roasted black soybean powder, consumed in a cookie matrix, on vascular function in healthy adults. Secondary measures included body composition and biomarkers of oxidative stress. Additionally, urinary and plasma polyphenol metabolites were measured in order to relate outcome measures to soybean polyphenol content. The authors found improvements in vascular function, systolic blood pressure, and some measures of oxidative stress, that latter which was negatively correlated with plasma and urine concentrations of select polyphenol metabolites. Although the range of measures made are interestingly, I have strong concerns about the analytical methods used (particularly statistics) and the reporting of results, and therefore feel that this paper requires major revisions before further review can take place. Abstract: Line 22: It appears vascular function is your primary outcome measure. Please indicate in your abstract how vascular function was measured. Line 28: Please identify which specific polyphenol metabolites are negatively correlatied with the oxidative stress marker 8-hydroxy-2'-deoxyguanosine. Introduction: Line 53-55: Please emphasize that physiological function you reference result from in vitro and animal models. Line 76: Was a test cookie used with the same formulation used in your previous study. If not, please provide a justification for changing the food matrix for delivering your chosen dose of black soybean powder. Materials and methods: Line 83-84: please remove "Participants were recruited that they had not any of the following" and replace with "Participants were excluded if they met any of the following exclusion criteria". Did the authors perform a sample size calculation for this study? Line 92: Please remove "finally" Line 94: How did they decide that a washout period of 4 weeks was sufficient for this trial? Line 95-99: Why did the authors choose to reduce the dose and intervention period of the trial? Line 106: Please provide a reference for the method used to measure the polyphenol content of the study cookies (what does "(Ref)" refer to?). Line 108: Do you mean antioxidant activity? Please spell out AAPH for its first instance (then you may use the abbreviation). Table 1A: Please elaborate on the specific ingredients used in the formulation of the study cookies (for example, what type of flour and syrup?). More detail is needed on the ingredient "indigestible dextrin". I calculate the total weight of the black soybean cookie to be 48.5 not 48.6 g (could be due to rounding). Table 1B: Was the nutritional composition of the study cookies determined experimentally? If so, I assume multiple replicates were used. Please provide a measure of variance around the values reported (e.g. standard deviation). If the nutritional composition was not determined experimentally, please describe how you obtained these values. I calculate the total caloric value of the placebo to be 201.7 kcal using the 4-4-9 rule. How did the authors derive the calorie values? Please provide this information in the table footnote. Also, did the authors measure the fibre content of the cookies? This would be an important measure for interpreting this study as dietary fibres are known to reduce blood pressue and improve endothelial function https://pubmed.ncbi.nlm.nih.gov/15668359/ Line 111-114: Please describe the "body composition meter" in more detail. How was it able to estimate BMR, bone mass, muscle mass, body fat, etc.? Line 123: Please provide more detail on how blood pressure was measured. Were measures taken after participants were allowed to rest for a period of time? Line 127: More detail is needed on the HRQOL. Please spell out this abbreviation in the secion heading. Section 2.8: I am unable to access the methodology for the extraction/quantification of polyphenols from the reference provided (#15). Could the authors please provide a description of these procedures or a copy of the methodology from the cited paper? Section 2.9 How were data evaluated for normality prior to testing? The tests that were run need to be described in detail. RESULTS Table 2: How was the polyphenol content of the test material determined? This needs to be described in the methodology. Please note that this is mentioned on line 107, but no actual reference is given. Were these determinations made in triplicate? Results should include a measure of variability. Line 168-170: Methodology for evaluating antioxidant activity in the test material is not provided (AAPH radical absorbance capacity) in methods section. What is the measure of variability for this measure? Were analyses performed in triplicate? Table 3: What statistical test were used to test whether there were differences between the intervention and control group? Given that this study is a crossover design, specific analytical approaches are required in order for this data to be valid. Figures: The quality of the figures could be improved by reducing the font size of the axis. Is it necessary to show figures for polyphenols for which concentration was "ND"? This seems unneccessary. Equol is spelled incorrectly as "equal". Line 235-246: Did the authors classify their study participants by equol-producing subtype? This could potentially yield some interesting findings. Other comments: I strongly recommend the authors seek the services of a copy editor to improve sentence structure, grammar, and to check for spelling errors in this manuscript. Was plasma antioxidant capacity measured? If so, the results of this measure should appear in the abstract. If not, why did the authors choose not to measure this when plasma was collected and the analytical methodology (AAPH radical absorbance capacity) was available?
Author Response
REVIEWER1
Thank you for your helpful comments and suggestions to improve our manuscript. We revised the manuscript according to your valuable comments. You will find our point-by-point answers below. Revised words and sentences are highlighted in the manuscript.
Comments
- Abstract: Line 22: It appears vascular function is your primary outcome measure. Please indicate in your abstract how vascular function was measured.
Answer: According to your comment, machinery to measure the vascular function was described in the abstract section.
- Line 28: Please identify which specific polyphenol metabolites are negatively correlated with the oxidative stress marker 8-hydroxy-2'-deoxyguanosine.
Answer: We appreciate your comment, because this is a very important point. Unfortunately, active compound was unclear in this study, the results from our previous study demonstrated that monomer to tetramer procyanidins from black soybean were reduced 8-OHdG and oxidative stress level as the same degree [9]. It was, therefore, suggested that these polyphenols were coordinately involved in the improvement of vascular function. It needs further study to clarify the active compound in future. We added sentences in the discussion section (pages 20-21, lines 359-364).
- Introduction: Line 53-55: Please emphasize that physiological function you reference result from in vitro and animal models.
Answer: According your comment, we added information of references what experimental model were used. We added information in the introduction section (page 4, lines 69, 70).
- Line 76: Was a test cookie used with the same formulation used in your previous study. If not, please provide a justification for changing the food matrix for delivering your chosen dose of black soybean powder.
Answer: Thank you very much for your comment. In this study, we want to evaluate whether short-term intake of black soybean with a less amount improved the vascular function in this trial. In the previous study [15], participants consumed 30 g of roasted black soybean as the test meal for 4 and 8 weeks and consumption of black soybean markedly improved the vascular function after 4 weeks. Therefore, in the present, we reduced the intake amount of black soybean to 20 g and decided the test period for 4 weeks. We added these reasons in the introduction and material and methods section (page 6, lines 92-94, and page7, lines 119-123).
- Materials and methods: Line 83-84: please remove "Participants were recruited that they had not any of the following" and replace with "Participants were excluded if they met any of the following exclusion criteria". Did the authors perform a sample size calculation for this study?
Answer: According to your comment, we removed the sentences (page 6, lines 105-106). In this study, we did not perform a sample size calculation.
- Line 92: Please remove "finally"
Answer: According to your comment, we deleted the word (page 7, line 116).
- Line 94: How did they decide that a washout period of 4 weeks was sufficient for this trial?
Answer: Thank you for your important question. Since polyphenols are recognized as xenobiotics, they are easily excreted into the urine and feces by 24hr [17]. In addition, many cross-over trials employed a 4-weeks washout period [18]. Thus, we also decided the 4-weeks washout period in the present study. These sentences were added in the material and methods section (page 7, lines 124-126).
- Line 95-99: Why did the authors choose to reduce the dose and intervention period of the trial?
Answer: Thank you for your comment, as you mentioned above in the Comment 4, please refer to the answer for the 4th comment.
- Line 106: Please provide a reference for the method used to measure the polyphenol content of the study cookies (what does "(Ref)" refer to?).
Answer: This is our mistake. We added a collect reference number and the method for the measurement of the polyphenol content in the material and methods section (page 8, lines 140-143).
- Line 108: Do you mean antioxidant activity? Please spell out AAPH for its first instance (then you may use the abbreviation).
According to your comment, AAPH was spell out to “AAPH; 2, 2'−Azobis (2−amidinopropane) dihydrochloride” (page 9, line 145-146 ). In addition, we added this word to the abbreviation list (page 3, line 45).
- Table 1A: Please elaborate on the specific ingredients used in the formulation of the study cookies (for example, what type of flour and syrup?). More detail is needed on the ingredient "indigestible dextrin". I calculate the total weight of the black soybean cookie to be 48.5 not 48.6 g (could be due to rounding).
Answer: Thank you for your important comment. In this study, used black soybean cultivar was “Hikariguro”, type of flour was wheat flour, and syrup was liquid sucrose. Indigestible dextrin was used to forming of cookie (Pine Fiber W, Matsutani Chemical Industry Co., Ltd, Itami, Japan). These detail information of the ingredients were added in the material and methods section and Table 1A (page 8, lines 134-138). As to the total weight of the black soybean cookie, we corrected the value in Table 1A as you pointed out.
- Table 1B: Was the nutritional composition of the study cookies determined experimentally? If so, I assume multiple replicates were used. Please provide a measure of variance around the values reported (e.g. standard deviation). If the nutritional composition was not determined experimentally, please describe how you obtained these values. I calculate the total caloric value of the placebo to be 201.7 kcal using the 4-4-9 rule. How did the authors derive the calorie values? Please provide this information in the table footnote. Also, did the authors measure the fibre content of the cookies? This would be an important measure for interpreting this study as dietary fibres are known to reduce blood pressure and improve endothelial function https://pubmed.ncbi.nlm.nih.gov/15668359/.
Answer: Thank you for your comment. In this study, we did not measure the nutritional composition by the experiment. The nutritional components were calculated the content of each nutritional component, which is included in the test cookies using the Standard Tables of Food Composition in JAPAN, 2015. We made a calculation mistake for the calorie values. We revised the total calorie value in Table 1B. The amount of dietary fiber was also calculated and added in the Table 1B. As your assumed, black soybean cookie contained about 6-fold higher amount of dietary fiber than placebo cookie. Components other than polyphenols in the black soybeans containing dietary fiber may also contribute to the observed effects. Actually, nutrient content of soybean and placebo cookies were quite different in the present study. We added some sentences in the discussion section (pages 21-22, lines 378-385).
- Line 111-114: Please describe the "body composition meter" in more detail. How was it able to estimate BMR, bone mass, muscle mass, body fat, etc.?
Answer: Thank you for your comment. The body composition, body weight, body mass index (BMI), body fat percentage, visceral fat percentage, biological age, basal metabolic rate, estimated bone mass, and muscle mass were measured by a Bioelectrical Impedance Analysis using a body composition meter (BC-610-PB, TANITA. Co., Ltd. Tokyo, Japan). We added this information in the material and methods section (page 9, line 152).
- Line 123: Please provide more detail on how blood pressure was measured. Were measures taken after participants were allowed to rest for a period of time?
Answer: Thank you for your comment. Participants took a rest about 10 min, and then blood pressure were measured using automated sphygmomanometer (HEM-9000AI, OMRON Corporation, Kyoto, Japan). The sentences were added in the material and methods section (page 9, lines 156-157).
- Line 127: More detail is needed on the HRQOL. Please spell out this abbreviation in the section heading.
Answer: Thank you for your comment. In this study, HRQOL was not changed by the intake of each cookie. This information was not important in the present study. Thus, this section was deleted.
- Section 2.8: I am unable to access the methodology for the extraction/quantification of polyphenols from the reference provided (#15). Could the authors please provide a description of these procedures or a copy of the methodology from the cited paper?
Answer: According to your comment, we changed the reference and added the methodology for the extraction and quantification of polyphenols in the material and methods section (page 11, lines 191-206 ).
- Section 2.9: How were data evaluated for normality prior to testing? The tests that were run need to be described in detail.
Answer: Thank you for comment for statistical analysis. We change the analysis method to the Wilcoxon signed-rank test. Since this is a nonparametric method, we do not need estimation of normality.
- Table 2: How was the polyphenol content of the test material determined? This needs to be described in the methodology. Please note that this is mentioned on line 107, but no actual reference is given. Were these determinations made in triplicate? Results should include a measure of variability.
Answer: Thank you for your comment. In Table 2, total polyphenol was measured triplicate. We added standard error in the Table. However, each polyphenol was a single analysis. Unfortunately, test cookies were not stored in the freezer. We could not perform in triplicate measurement. Please understand our situation. We added the methodology of the polyphenol content in the test material (page 8, lines 140-143).
- Line 168-170: Methodology for evaluating antioxidant activity in the test material is not provided (AAPH radical absorbance capacity) in the methods section. What is the measure of variability for this measure? Were analyses performed in triplicate?
Answer: Yes, the measurement was triplicate. According to your comment, we added the methodology of the antioxidant ability in the test material (page 13, line 233).
- Table 3: What statistical test were used to test whether there were differences between the intervention and control group? Given that this study is a crossover design, specific analytical approaches are required in order for this data to be valid.
Answer: Thank you for you very important comment. We mistake the choice of the statistical analysis. We re-analyzed the all data again with the Wilcoxon signed-rank test. Before the first and second (after washout period) trials. We confirm that there was no statistical difference between the groups. Thus, we merged data for the first and second trials and analyzed statistical analysis between the intervention and control groups.
- Figures: The quality of the figures could be improved by reducing the font size of the axis. Is it necessary to show figures for polyphenols for which concentration was "ND"? This seems unneccessary. Equol is spelled incorrectly as "equal".
Answer: c The spell of equol was corrected throughout the manuscript.
- Line 235-246: Did the authors classify their study participants by equal-producing subtype? This could potentially yield some interesting findings.
Answer: Thank you for your comment. In the present study, we measured only S type isomer of equol. Because S-equol is major metabolite of isoflavone in human and R type isomer is not almost produce.
- Other comments: I strongly recommend the authors seek the services of a copy editor to improve sentence structure, grammar, and to check for spelling errors in this manuscript.
Answer: We checked English throughout the manuscript. In addition, we ask to request the English editing of our manuscript to the Editorial Office.
- Was plasma antioxidant capacity measured? If so, the results of this measure should appear in the abstract. If not, why did the authors choose not to measure this when plasma was collected and the analytical methodology (AAPH radical absorbance capacity) was available?
Answer: Thank you for your very important comment. In this study, we evaluated the antioxidant ability by 8OHdG, HEL and MPO, but not AAPH in the plasma and urine. Because, we focused on the biological function after intake of black soybean, a substance from the organism and its product were important. In addition, AAPH radical absorbance capacity is difficult to applicable for measurement of in vivo antioxidant activity, though this method is well-used for measurement of in vitro antioxidant activity due to simple and convenient method.
Reviewer 2 Report
To explore the impact of black soybeans on vascular function and blood pressure, the autors used a randomized, single blinded, placebo controled, crossover trial in 23 healthy adults, aged 30 – 60, evidencing a vascular age higher than their chronological age.
The study is well described but the English needs to be thoroughly revised and typos corrected. In addition, the clarity of the abstract can be further improved. In particular in the conclusion, the causal relationship described between improved vascular function and reduced oxidative stress is inappropriate (because not demonstrated in the study) and should be tempered.
Vascular age is a key criteria of inclusion in the study. It also supports one of the main results showing the beneficial effect of black soybeans. Its calculation and validation should thus be clearly explained, with associated references.
Line 67-69 : please, specify that results were obtained in vitro and in mice.
Lines 88-89 : add a reference to paragraph 2.5 for the calculation of the vascular age.
Line 113 : was biological age determined using the body composition meter ?! Was basal metabolic rate estimated or measured by indirect calorimetry ? Same question for the body composition, which technique is used to determine /estimate bone, muscle, fat masses ?
Table 1B : Protein and carbohydrate contents of cookies are quite different between the two treatements. PLease, elaborate this point in the discussion.
Line 126 : please define HRQOL
Table 5 : could other markers be used to support data on NO, such as plasma concentration in arginine and citrulline ?
Figure 1 : Please delete all ND figures and keep only those with measurable concentrations.
In the discussion, please elaborate about the physiological meaning of a 0.7 point reduction in systolic pressure in healthy normotensive individiuals. What impact could be expected in patients with hypertension ?
Author Response
We appreciate your vulnerable comments and suggestions. According to your comments and suggestions, we revised the manuscript and added some discussion. Below you will find our point-by-point answers. Revised words and sentences are highlighted in the manuscript.
- Line 67-69 : please, specify that results were obtained in vitro and in mice.
Answer: According your comment, we added information of references what experimental model were used. We added information in the introduction section (page, line).
- Lines 88-89 : add a reference to paragraph 2.5 for the calculation of the vascular age.
Answer: Thank you for pointed out our mistake. We added a collect reference number and the method for measurement of the polyphenol content in the material and methods section (page 4, lines 69-70).
- Line 113 : was biological age determined using the body composition meter ?! Was basal metabolic rate estimated or measured by indirect calorimetry ? Same question for the body composition, which technique is used to determine /estimate bone, muscle, fat masses ?
Answer: Thank you for your comment. The body composition, body weight, body mass index (BMI), body fat percentage, visceral fat percentage, biological age, basal metabolic rate, estimated bone mass, and muscle mass were measured by a Bioelectrical Impedance Analysis using a body composition meter (BC-610-PB, TANITA. Co., Ltd. Tokyo, Japan). We added this information in the material and methods section (page 9, line 152).
- Table 1B : Protein and carbohydrate contents of cookies are quite different between the two treatements. Please, elaborate this point in the discussion.
Answer: Thank you for your very important comment. As your assumed, components other than polyphenols in the black soybeans may also contribute to the observed effects. Actually, nutrient content of soybean and placebo cookies were quite different in the present study. We added discussion about this issue in the discussion section (pages 21-22, lines 378-387 ).
- Line 126 : please define HRQOL
Answer: Thank you for your comment. In this study, HRQOL was not changed by the intake of each cookie. This information was not important in the present study. Thus, this section was deleted.
- Table 5 : could other markers be used to support data on NO, such as plasma concentration in arginine and citrulline ?
Answer: Thank you for your important comment. Unfortunately, in the present study, we measured only the NO level as a vascular function marker, because the sample volume was limited.
- Figure 1 : Please delete all ND figures and keep only those with measurable concentrations.
Answer: According to your comment, we deleted the panels of not detectable polyphenols from the figures.
- In the discussion, please elaborate about the physiological meaning of a 0.7 point reduction in systolic pressure in healthy normotensive individuals. What impact could be expected in patients with hypertension ?
Answer: As you pointed out, our results provide impact information for the hypertension patients. However, in Japan, functional foods are distinct from the medicines. Therefore, the function of food materials should be evaluated in healthy subjects, but not patients. Thus, we cannot conduct the patient human study. Please understand our situation in Japan. Nonetheless, black soybean is a nutritious and functional food that should be recommended for daily consumption. This sentence was added in the last of discussion section (page 22, lines 386-387).
Round 2
Reviewer 1 Report
Thank you for addressing the majority of the comments initially made. I believe more work is needed to address the original concerns I outlined. I have provided numbered responses to the authors' comments in attempt to clarify my concerns over how this research has been presented. Response to Authors' Comments 1. Answer: According to your comment, machinery to measure the vascular function was described in the abstract section. Response: Remove quotations from "Pulse Analyzer" and include a trademark or registered symbol if this is the name of a branded instrument. 2. Answer: We appreciate your comment, because this is a very important point. Unfortunately, active compound was unclear in this study, the results from our previous study demonstrated that monomer to tetramer procyanidins from black soybean were reduced 8-OHdG and oxidative stress level as the same degree [9]. It was, therefore, suggested that these polyphenols were coordinately involved in the improvement of vascular function. It needs further study to clarify the active compound in future. We added sentences in the discussion section (pages 20-21, lines 359-364). Response: You must clarify how polyphenol concentrations in plasma and urine were measured by, for example, including the statement "...by increasing concentration of polyphenols derived from black soybean in plasma and urine, as measured by _____ ". 4. Thank you very much for your comment. In this study, we want to evaluate whether short-term intake of black soybean with a less amount improved the vascular function in this trial. In the previous study [15], participants consumed 30 g of roasted black soybean as the test meal for 4 and 8 weeks and consumption of black soybean markedly improved the vascular function after 4 weeks. Therefore, in the present, we reduced the intake amount of black soybean to 20 g and decided the test period for 4 weeks. We added these reasons in the introduction and material and methods section (page 6, lines 92-94, and page7, lines 119-123). Response: The authors have not answered my question. Was the test meal in the previous study a cookie that incorporated 30 g roasted soybean, or was it a different food? 5. Answer: According to your comment, we removed the sentences (page 6, lines 105-106). In this study, we did not perform a sample size calculation. Response: Please indicate in the materials and methods section that a sample size calculation was not made for this study. This is an important piece of information that enables to critically evaluate the quality of a clinical trial study. 10. According to your comment, AAPH was spell out to “AAPH; 2, 2'−Azobis (2−amidinopropane) dihydrochloride” (page 9, line 145-146 ). In addition, we added this word to the abbreviation list (page 3, line 45). Response: The authors have left the term "antioxidant ability" in, which is incorrect terminology. Should say "antioxidant activity". or "antioxidant capacity". 12. Answer: Thank you for your comment. In this study, we did not measure the nutritional composition by the experiment. The nutritional components were calculated the content of each nutritional component, which is included in the test cookies using the Standard Tables of Food Composition in JAPAN, 2015. We made a calculation mistake for the calorie values. We revised the total calorie value in Table 1B. The amount of dietary fiber was also calculated and added in the Table 1B. As your assumed, black soybean cookie contained about 6-fold higher amount of dietary fiber than placebo cookie. Components other than polyphenols in the black soybeans containing dietary fiber may also contribute to the observed effects. Actually, nutrient content of soybean and placebo cookies were quite different in the present study. We added some sentences in the discussion section (pages 21-22, lines 378-385). Response: The recipe shown in Table1A and the nutritional value in Table 1B are not making sense to me. Please double-check these values, particularly the fibre content. I believe 20 g of soy flour should contain ~3.5 g fibre. Was the roasted black soybean powder de-fatted or full-fat? The authors also incorrectly state 54.2 g of sodium are included in the soy cookie, when it should be 54.2 mg. I strongly urge the authors to review their data more carefully. 15. Answer: Thank you for your comment. In this study, HRQOL was not changed by the intake of each cookie. This information was not important in the present study. Thus, this section was deleted. Response: The HRQOL abbreviation remains in the Abbreviations section at the bottom of page 1. If this information was deemed not important, why was it measured in the first place? A nonsignificant finding is still worth reporting and it is rather unethical to omit. 17. Answer: Thank you for comment for statistical analysis. We change the analysis method to the Wilcoxon signed-rank test. Since this is a nonparametric method, we do not need estimation of normality. Response: The authors seem to misunderstand what statistical techniques are appropriate for the analysis of crossover clinical trial designs. A Wilcoxon signed-rank test is not appropriate. I recommend the authors consult the following papers for guidance: https://trialsjournal.biomedcentral.com/articles/10.1186/1745-6215-10-27 https://journals.plos.org/plosone/article?id=10.1371/journal.pone.0133023 18. Answer: Thank you for your comment. In Table 2, total polyphenol was measured triplicate. We added standard error in the Table. However, each polyphenol was a single analysis. Unfortunately, test cookies were not stored in the freezer. We could not perform in triplicate measurement. Please understand our situation. We added the methodology of the polyphenol content in the test material (page 8, lines 140-143). Response: Are the values for total polyphenol content means +/- standard deviation or standard error? In the statistics section, the authors indicate that values are expressed as mean +/- standard deviation. 20. Answer: Thank you for you very important comment. We mistake the choice of the statistical analysis. We re-analyzed the all data again with the Wilcoxon signed-rank test. Before the first and second (after washout period) trials. We confirm that there was no statistical difference between the groups. Thus, we merged data for the first and second trials and analyzed statistical analysis between the intervention and control groups. Response: This approach is incorrect. The authors must indicate when findings are insignificant. Omitting these results and then re-analyzing the data in attempt to find significant findings is a form of "p-value fishing" and is misleading. I strongly urge the authors to carefully report all the steps they took to analyze their data and report all findings. This enables the reader to critically evaluate the research being presented. 22. Answer: Thank you for your comment. In the present study, we measured only S type isomer of equol. Because S-equol is major metabolite of isoflavone in human and R type isomer is not almost produce. Response: S-equol is indeed a major metabolite of the isoflavone daidzin, and has strong biological activity, but it is only produced by some members of the population. Therefore, it has been hypothesized that "equol producers" may receive health benefits from consuming soy that nonproducers do not. It is valuable to the reader for the authors to define their study population on this basis if equol measurements have been made. https://academic.oup.com/jn/article/136/8/2188/4664801 https://academic.oup.com/jn/article/132/12/3577/4712130 23. Answer: We checked English throughout the manuscript. In addition, we ask to request the English editing of our manuscript to the Editorial Office. Response: Additional improvements are still needed.
Author Response
To the Reviewer 1:
Thank you for addressing the majority of the comments initially made. I believe more work is needed to address the original concerns I outlined. I have provided numbered responses to the authors' comments in attempt to clarify my concerns over how this research has been presented. Response to Authors' Comments
Thank you again for reviewing the manuscript and providing the comments. Followings are our response to the comments by a point-by-point style.
1. Answer: According to your comment, machinery to measure the vascular function was described in the abstract section.
Response: Remove quotations from "Pulse Analyzer" and include a trademark or registered symbol if this is the name of a branded instrument.
According to your comment, quotations were deleted and added ® as the Registered Trademark on Page 1, Line 27; Page 4, Line 161; and Page 4, line 169.
2. Answer: We appreciate your comment, because this is a very important point. Unfortunately, active compound was unclear in this study, the results from our previous study demonstrated that monomer to tetramer procyanidins from black soybean were reduced 8-OHdG and oxidative stress level as the same degree [9]. It was, therefore, suggested that these polyphenols were coordinately involved in the improvement of vascular function. It needs further study to clarify the active compound in future. We added sentences in the discussion section (pages 20-21, lines 359-364).
Response: You must clarify how polyphenol concentrations in plasma and urine were measured by, for example, including the statement "...by increasing concentration of polyphenols derived from black soybean in plasma and urine, as measured by _____ ".
Method for the measurement of polyphenols was written in the Materials and Methods section (2.8. Extraction and quantification of polyphenols). Please confirm it. To clarify the used method in the result section, we added the words "...by HPLC... " at the first sentence in the "3.6. Polyphenol concentrations in the plasma and urine" sub-section (Page 12, Line 355) and in the third paragraph of the Discussion section (Page 18, Line 437).
3. Thank you very much for your comment. In this study, we want to evaluate whether short-term intake of black soybean with a less amount improved the vascular function in this trial. In the previous study [15], participants consumed 30 g of roasted black soybean as the test meal for 4 and 8 weeks and consumption of black soybean markedly improved the vascular function after 4 weeks. Therefore, in the present, we reduced the intake amount of black soybean to 20 g and decided the test period for 4 weeks. We added these reasons in the introduction and material and methods section (page 6, lines 92-94, and page7, lines 119-123).
Response: The authors have not answered my question. Was the test meal in the previous study a cookie that incorporated 30 g roasted soybean, or was it a different food?
We are sorry for confusing you. We used whole grain of roasted black soybeans in the previous trial, while a cookie containing powdered roasted black soybeans was used in this trial. To avoid confusion, we added the words, "(whole grain)" on Page 3, Lines 100-101.
4. Answer: According to your comment, we removed the sentences (page 6, lines 105-106). In this study, we did not perform a sample size calculation. Response: Please indicate in the materials and methods section that a sample size calculation was not made for this study. This is an important piece of information that enables to critically evaluate the quality of a clinical trial study. 10. According to your comment, AAPH was spell out to “AAPH; 2, 2'−Azobis (2−amidinopropane) dihydrochloride” (page 9, line 145-146 ). In addition, we added this word to the abbreviation list (page 3, line 45).
Response: The authors have left the term "antioxidant ability" in, which is incorrect terminology. Should say "antioxidant activity". or "antioxidant capacity".
According to your comment, "antioxidant ability" was changed to "antioxidant activity" throughout the manuscript.
12. Answer: Thank you for your comment. In this study, we did not measure the nutritional composition by the experiment. The nutritional components were calculated the content of each nutritional component, which is included in the test cookies using the Standard Tables of Food Composition in JAPAN, 2015. We made a calculation mistake for the calorie values. We revised the total calorie value in Table 1B. The amount of dietary fiber was also calculated and added in the Table 1B. As your assumed, black soybean cookie contained about 6-fold higher amount of dietary fiber than placebo cookie. Components other than polyphenols in the black soybeans containing dietary fiber may also contribute to the observed effects. Actually, nutrient content of soybean and placebo cookies were quite different in the present study. We added some sentences in the discussion section (pages 21-22, lines 378-385).
Response: The recipe shown in Table1A and the nutritional value in Table 1B are not making sense to me. Please double-check these values, particularly the fibre content. I believe 20 g of soy flour should contain ~3.5 g fibre. Was the roasted black soybean powder de-fatted or full-fat? The authors also incorrectly state 54.2 g of sodium are included in the soy cookie, when it should be 54.2 mg. I strongly urge the authors to review their data more carefully.
Thank you for your careful review of the data. As you pointed out, we made a calculation mistake. In Table 1B, dietary fiber content and unit of sodium were corrected.
15. Answer: Thank you for your comment. In this study, HRQOL was not changed by the intake of each cookie. This information was not important in the present study. Thus, this section was deleted.
Response: The HRQOL abbreviation remains in the Abbreviations section at the bottom of page 1. If this information was deemed not important, why was it measured in the first place? A nonsignificant finding is still worth reporting and it is rather unethical to omit.
According you comment, explanations for HRQOL was added again in the manuscript (Page 1, Line 40 – Page 2, Line 41; Page 3, Lines 107-108; Page 5, Lines 211-219; and Page 6, Lines 266-267.
17. Answer: Thank you for comment for statistical analysis. We change the analysis method to the Wilcoxon signed-rank test. Since this is a nonparametric method, we do not need estimation of normality.
Response: The authors seem to misunderstand what statistical techniques are appropriate for the analysis of crossover clinical trial designs. A Wilcoxon signed-rank test is not appropriate. I recommend the authors consult the following papers for guidance: https://journals.plos.org/plosone/article?id=10.1371/journal.pone.0133023
Thank you for providing a paper indicating the guidance for the analysis of crossover clinical trial. According to your kind suggestion, we added supporting data in Tables S1-S5 for Tables 3-5 and Figures 1 and 2, respectively. As to the statistical analysis method, we change the analysis method from a Wilcoxon signed-rank test to a Welch test, which is a non-parametric t-test.
18. Answer: Thank you for your comment. In Table 2, total polyphenol was measured triplicate. We added standard error in the Table. However, each polyphenol was a single analysis. Unfortunately, test cookies were not stored in the freezer. We could not perform in triplicate measurement. Please understand our situation. We added the methodology of the polyphenol content in the test material (page 8, lines 140-143).
Response: Are the values for total polyphenol content means +/- standard deviation or standard error? In the statistics section, the authors indicate that values are expressed as mean +/- standard deviation.
We added a following sentence in the statistics section: Data are expressed as the means ± standard deviation (Page 5, Line 221).
20. Answer: Thank you for you very important comment. We mistake the choice of the statistical analysis. We re-analyzed the all data again with the Wilcoxon signed-rank test. Before the first and second (after washout period) trials. We confirm that there was no statistical difference between the groups. Thus, we merged data for the first and second trials and analyzed statistical analysis between the intervention and control groups.
Response: This approach is incorrect. The authors must indicate when findings are insignificant. Omitting these results and then re-analyzing the data in attempt to find significant findings is a form of "p-value fishing" and is misleading. I strongly urge the authors to carefully report all the steps they took to analyze their data and report all findings. This enables the reader to critically evaluate the research being presented.
Please refer to our response against your specific comment #7 (previous comment #17). We added supporting data in Tables S1-S5 for Tables 3-5 and Figures 1 and 2, respectively. As to the statistical analysis method, we change the analysis method from a Wilcoxon signed-rank test to a Welch test, which is a non-parametric t-test.
22. Answer: Thank you for your comment. In the present study, we measured only S type isomer of equol. Because S-equol is major metabolite of isoflavone in human and R type isomer is not almost produce.
Response: S-equol is indeed a major metabolite of the isoflavone daidzin, and has strong biological activity, but it is only produced by some members of the population. Therefore, it has been hypothesized that "equol producers" may receive health benefits from consuming soy that nonproducers do not. It is valuable to the reader for the authors to define their study population on this basis if equol measurements have been made. https://academic.oup.com/jn/article/136/8/2188/4664801 https://academic.oup.com/jn/article/132/12/3577/4712130
Thank you very much for your important comment. We calculated concentration range of equol in the plasma and urine. Results were added in the Result section on Page 17, Lines 400-406. Contribution of equol to the vascular function was discussed on Page 18, Lines 449-452.
23. Answer: We checked English throughout the manuscript. In addition, we ask to request the English editing of our manuscript to the Editorial Office.
Response: Additional improvements are still needed.
We checked English again and revised the manuscript.
Reviewer 2 Report
The authors have appropriately answered to reviewers' comments.
Author Response
To the Reviewer 2:
Thank you again for reviewing the manuscript. Followings are our response to the comments by a point-by-point style.
2. Lines 88-89 : add a reference to paragraph 2.5 for the calculation of the vascular age. Authors’ answer: Thank you for pointed out our mistake. We added a collect reference number and the method for measurement of the polyphenol content in the material and methods section (page 4, lines 69-70).
The authors’ answer does not correctly respond the above comment. The reviewer 2 asked the authors to add a reference for the calculation of the vascular age on that instrument. I also ask the authors to add information on the accuracy and validity of that measurement. What does “vascular age” reflect in terms of vascular function?
We are sorry for incorrect response. The explanation for the calculation of vascular age, explanation sentence with three references was added as follows: "Vascular age was calculated from second derivative of photoplethysmogram aging index which is [(b-c-d-e)/a] wave ratio [19-21]." (Page 4, Lines 166-167). Since the second derivative of photoplethysmogram signal is called APG, the accuracy and validity of this study is the same as the results from the commonly used photoplethysmogram device. "Vascular age" is one of the markers for the vascular function, because aging leads to the vascular dysfunction as mentioned in the Introduction section (Page 2, Lines 51-52).
Reviewer 2’ comment: 8. In the discussion, please elaborate about the physiological meaning of a 0.7 point reduction in systolic pressure in healthy normotensive individuals. What impact could be expected in patients with hypertension? Authors’ answer: As you pointed out, our results provide impact information for the hypertension patients. However, in Japan, functional foods are distinct from the medicines. Therefore, the function of food materials should be evaluated in healthy subjects, but not patients. Thus, we cannot conduct the patient human study. Please understand our situation in Japan. Nonetheless, black soybean is a nutritious and functional food that should be recommended for daily consumption. This sentence was added in the last of discussion section (page 22, lines 386- 387).
The reviewer questioned that “Does a 0.7 point (probably meaning 7 mmHg) reduction in systolic blood pressure in participants really have clinical significance?”. As pointed by the reviewer 1, the authors should revise the statistical analyses and re-evaluate the results of this study based on their statistics results. I think the added sentence “Nonetheless, black soybean is a nutritious and functional food that should be recommended for daily consumption.” is not scientific.
In this study, central blood pressure tended to decrease at the end of the trial without statistical significance (Page 8 Lines 296-298 and Table 4), in addition to systolic blood pressure, which was significantly decreased at the end of trial compared to that at the start of trial. After the re-evaluation of the results of blood pressure based on the statistics results according to the comments from the Reviewer #1, significant difference of systolic blood pressure was observed between the black soybean and placebo groups at the end of trial. Sentence was added in the text from Page 8, Lines 294-299. From these results, we believe that reduction in systolic blood pressure in participants is meaningful. Since functional foods are distinct from the medicines in Japan, we cannot enroll the patients for human trial. Please understand our situation.
As to the sentence "Nonetheless, black soybean is a nutritious and functional food that should be recommended for daily consumption. ", we revised this sentence to "Therefore, black soybean is an attractive functional food for prevention and/or improvement of vascular function. Daily intake of black soybean may contribute to maintain human health." (Page 18, Line 474 – Page 19, Line 477)
Overall comment: Please indicate the page and line number in the final draft (as Nutrients format) for the corrections.
All corrections were written by red characters and highlighted with yellow color. The page and line number(s) were indicated in this response sheet.